# Toughening Mechanism Analysis of Recycled Rubber-Based Composites Reinforced with Glass Bubbles, Glass Fibers and Alumina Fibers

**DOI:** 10.3390/polym13234215

**Published:** 2021-12-01

**Authors:** Gamze Cakir Kabakci, Ozgur Aslan, Emin Bayraktar

**Affiliations:** 1Department of Mechanical Engineering, Atilim University, Ankara 06830, Turkey; gamze.cakir@atilim.edu.tr (G.C.K.); ozgur.aslan@atilim.edu.tr (O.A.); 2ISAE-Supmeca-Paris, School of Mechanical and Manufacturing Engineering, 93400 Paris, France

**Keywords:** toughening mechanisms, recycled rubber, glass bubble microspheres, Halpin-Tsai, scanning electron microscopy

## Abstract

Recycling of materials attracts considerable attention around the world due to environmental and economic concerns. Recycled rubber is one of the most commonly used recyclable materials in a number of industries, including automotive and aeronautic because of their low weight and cost efficiency. In this research, devulcanized recycled rubber-based composites are designed with glass bubble microsphere, short glass fiber, aluminum chip and fine gamma alumina fiber (γ-Al_2_O_3_) reinforcements. After the determination of the reinforcements with matrix, bending strength and fracture characteristics of the composite are investigated by three-point bending (3PB) tests. Halpin–Tsai homogenization model is adapted to the rubber-based composites to estimate the moduli of the composites. Furthermore, the relevant toughening mechanisms for the most suitable reinforcements are analyzed and stress intensity factor, K_Ic_ and critical energy release rate, G_Ic_ in mode I are determined by 3PB test with single edge notch specimens. In addition, 3PB tests are simulated by finite element analysis and the results are compared with the experimental results. Microstructural and fracture surfaces analysis are carried out by means of scanning electron microscopy (SEM). Mechanical test results show that the reinforcement with glass bubbles, aluminum oxide ceramic fibers and aluminum chips generally increase the fracture toughness of the composites.

## 1. Introduction

Recycling of materials attracts considerable attention worldwide due to raw material costs, shrinking resources as well as environmental issues. Among the recyclable materials, recycled rubber is a widely used material in many industries such as aeronautic, automotive and transportation as they can be suitable for applications where high toughness, high resistance to impact, low cost and lightweight structures are desired. For instance, reduction of overall cost and also mass reduction to lower fuel consumption and CO_2_ emissions of an aircraft is desirable by aeronautic companies. Therefore, development of low-cost and lightweight materials to be used in the manufacture constitutes an important task. They are reinforced with microscale particles to develop physical and mechanical properties of the recycled rubber, such as fracture toughness, resistance to impact. Glass bubbles, alumina fiber, nanosilica, graphene nanoplatelets, etc. are used as reinforcement of composites in the literature. Glass bubbles are an alternative to commonly used inorganic fillers, such as silica, calcium carbonate and talc, for many uses of reinforced rubber composites [1]. They have a high strength to density ratio for use in demanding polymer processing operations for various applications. A perfectly spherical shape with an aspect ratio of one makes glass bubbles efficient volume fillers and they, when used with other reinforcing fillers in an optimized formulation, can provide excellent weight reduction, performance, processing and dimensional stability characteristics [1]. Alumina fibers and nanosilica particles improve fracture toughness, resin stiffness and wear resistance. Alumina has good thermal conductivity, inertness to most acids and alkalis, high absorption capacity, thermal stability and electrical insulation [2]. For instance, interfacial strength and elastic modulus can improve strength and fracture mechanisms. Moreover, alumina fibers have high compressive strength and shear properties [3]. Then, graphene (GnPs) is an alternative nano-scale material with high static and dynamic mechanical and physical properties. It has a two-dimensional honeycomb structure and high elasticity modulus and mechanical strength [4]. Furthermore, epoxy resin is used as binder between rubber and reinforcements. During the curing of epoxy resin, a 3D network is created between epoxy molecules and reactive groups of the curing agent. The level of cross-linking has a significant effect on the mechanical properties. A lower crosslink density decreases the storage modulus but it improves the fracture toughness. Moreover, devulcanization is realized to improve the interaction and the adhesion between waste rubber and the matrix. Furthermore, devulcanization of cured rubber ensures the reversion of the crosslinking process, which leads to obtain recycled material with properties very close to the original material [5]. In the literature, there are several studies on design and manufacturing of recycled rubber composites. Irez, Zambelis and Bayraktar [6] have studied the fracture behavior and damage analyses of recycled rubber modified epoxy-based composites reinforced with alumina fiber. They have used recycled fresh scrap rubbers, epoxy resin and alumina fibers and conducted three-point bending tests to investigate fundamental mechanical characteristics. Mechanical test results have shown that the reinforcement with alumina fibers improves the fracture toughness of the composites. Moreover, numerical studies have indicated that energy release rate shows some variations along the specimen thickness. It is stated that recycled rubber composites are promising candidates to manufacture the various components in automotive industry. Another study on recycled rubber composites has been performed by Irez, Bayraktar and Miskioglu [7]. Recycled and recycled rubber modified epoxy-based composites reinforced with nano-magnetic iron oxide have been designed. Nano-magnetic iron oxide, Fe_3_O_4_, is added to the matrix as reinforcement elements in different percentages. Nickel and aluminum are also added as auxiliary additional elements. All of the tests, including scanning electron microscopy, nanoindentation and static (3PB) test, revealed a combined effect of toughening mechanisms, resulting in high-strength, lightweight and low-cost composites based on rubber modified epoxy composites reinforced with nano magnetic iron oxide and auxiliary fine nickel and nano aluminum powders. The authors also have studied the fracture toughness analysis of epoxy-recycled rubber-based composites reinforced with graphene nanoplatelets for structural applications in automotive and aeronautics [8]. After manufacturing the composites, their bending strength and fracture characteristics have been investigated by 3PB tests. Halpin–Tsai homogenization adapted to composites containing GnPs has been used to estimate the moduli of the composites, and satisfactory agreement with the 3PB test results was observed. In addition, 3PB tests have been simulated by finite element method incorporating the Halpin–Tsai homogenization, and the resulting stress–strain curves have been compared with the experimental results. Mechanical test results have shown that the reinforcement with GnPs generally increases the modulus of elasticity as well as the fracture toughness of these novel composites. The majority of the studies on recycled rubber composites show that recycled rubber reinforced with nano-scale particles lead to develop physical and mechanical properties of the structures and also provide low-cost and lightweight composites for several application areas [9,10,11,12,13,14].

In this study, low cost devulcanized recycled rubber based composites have been designed with glass bubble microspheres, short glass fibers and fine gamma alumina fiber (γ-Al_2_O_3_) reinforcements. Then, the relevant toughening mechanisms for the most suitable reinforcements are analyzed in detail and certain mechanical and physical properties are determined by fracture toughness tests. Microstructural and fracture surfaces analysis are carried out by means of Scanning Electron Microscopy (SEM).

## 2. Experimental Procedure

Low-cost recycled rubber-based composites have been reinforced with glass bubble microspheres, short glass fibers, very fine aluminum chips (Al-chips) and fine gamma alumina fibers (γ-Al_2_O_3_) reinforcements containing minor reinforcements such as nanosilica (SiO_2_) and hexagonal-boron nitride (h-BN) to provide multifunctionality to the composites. In this study, glass bubbles are preferred as major reinforcement as they lead to perfect weight reduction, performance, processing and dimensional stability characteristics when used with other reinforcing fillers [15]. Furthermore, glass bubbles have a thin-walled hollow spherical shape, soda-lime-borosilicate glass composition and have high strength to density ratio. The hollow glass bubbles have a typical density of 0.12 g/cm^3^, average diameter of 30 μm and modulus of elasticity of 3546 MPa. A perfect spherical shape with an aspect ratio of one makes them efficient volume fillers as they can be incorporated at very high volume loads without increasing the viscosity to unacceptable levels for further polymer operations. Moreover, recycled rubber is supplied by the sports equipment manufacturers as fresh scrap collected directly from the production line. The average diameter of the rubber particles is 50 μm, the modulus of elasticity is 10 MPa, the elongation at break is 80–100%, the hardness is 37–40 “Shore A” and the density is 1 g/cm^3^. In addition, recycled glass fibers are included as one of the major reinforcements in this research. Their typical density is 2.45 g/cm^3^ and they also have high strength with low density. Moreover, alumina fiber and nanosilica generally improve fracture toughness, resin stiffness and wear resistance and have high compressive strength and shear properties [3] and alumina has also good thermal conductivity and it is inexpensive, non-toxic and highly abrasive [9,16,17]. Furthermore, epoxy resin is used as binder between recycled rubber and reinforcements. The level of cross-linking has a significant effect on the mechanical properties [17]. The toughness improvement in hybrid rubber based composites can increase the plastic zone in the structure and this leads the rubber based composites to dissipate additional fracture energy. For this reason, the mixture of composite material is chosen as 80 weight percent (wt%) of recycled rubber and 20 wt% of epoxy. For manufacturing of recycled rubber-based composites, a good quality of bonding between matrix and reinforcements is required, however recycled rubbers do not carry any free chains to form new bonding with epoxy resin because of former vulcanization process. In the present paper, we tried to heat with microwave oven during 4 min to be devulcanized it. During this process, we attempt to break sulfur links and partially new other links are generated, then the flowing capacity and interaction of recycled rubbers with other substances are increased. In the literature, some works have already indicated that devulcanization cannot be achieved completely. As we have shown in our former works [6,7,8,18,19,20,21], we have only carried out 10–12% of devulcanization. After determination of the reinforcements with matrix in wt%, a special process has been applied to complete successfully the manufacturing of the composites. This process involves silanization of the recycled rubber and devulcanization before blending it with epoxy resin and reinforcement. With this simple process, we try to improve only partially good liaison between recycled rubber matrix with the reinforcement. It means that we want to obtain a chemical bonding diffusion with different reinforcement. Chemical compositions of the composites in weight percentage that are used to prepare the specimens are presented in the Table 1.

After blending of this mixture, they have been milled during 4 h by adding “Zn-Stearate” as lubricant for homogenous distribution. For manufacturing of the composites, hot compaction has been carried out at 180–200 °C during 20 min under the pressure of 6 tons after that the specimens produced with a diameter of 50 mm. Final rectangular-shaped specimens have been cut for 3PB tests (ASTM STP 856) called Single Edge Notch Bending (SENB) test specimens to calculate fracture toughness and fracture energy values of these composites.

The most common flexural test for composite materials is the three-point bending test. Flexural strength, elasticity modulus in bending and strain are obtained from the test results. Three-point bending tests are carried out according to the ASTM D790 standards. Single Edge Notch Bending (SENB) test specimen (Figure 1) is placed to the universal testing machine and force is applied until its failure. In this figure, W is the depth and B is the width of the specimen. In addition, fracture toughness indicators such as critical stress intensity factor for mode I and critical strain energy release rate are investigated with notched specimens. During the tests, crosshead speed is selected between 1–2 mm/min and flexural strength and strain are obtained from the test results.

After the realization of bending tests, fracture surface analyses have been conducted by using Scanning Electron Microscope (SEM) on the damaged specimens. Microstructural analyses have also been carried out on the polished surfaces of these specimens (distribution of the particles by mapping and chemical analyses by EDS (Electro Discharge Spectrum)). All of the microhardness measurements have been carried out by using Shore D hardness tests to determine the depth of penetration.

## 3. Results and Discussions

### 3.1. Physical Properties, Microstructural Observations

For the Shore-D hardness test, at least five measurements have been obtained from the different points on the specimen surface, and the average values are presented in the Table 2. The fillers used as reinforcement generally enhance the surface hardness. It is observed that increasing the amount of aluminum chips increases the surface hardness whereas the amount of the glass bubbles and ceramic fibers has small effect on the surface hardness. The unexpected results may be due to inhomogeneous distribution of the fillers on the measurement point. For this reason, the results give a range for the hardness on the thickness section rather than exact values.

In terms of order of level of the hardness values, they are well beyond the hardness values of conventional rubber-based components that is a constructive effect of the reinforcement particles. In fact, these surface hardness values should be increased homogeneously on the entire of the surface for these novel designs of the rubber-based composites because their contact area with other components should be higher toughness for the tribological cases (surface damage due to surface friction between two different surfaces).

The microstructure analyses of the composites have been carried out by using SEM and a mapping analysis has also been done on the microstructures of the some of the polished specimens for observing the distribution of the reinforcement in the matrix. Figure 2 gives the details on the microstructure evolution with different reinforcements for seven composites after sectioning and polishing. One may observe a homogenous distribution of the reinforcement in the matrix with some small agglomeration of the fine powders used as reinforcements in the matrix such as glass bubbles. The decrease in strain efficiency of composites may be due to the agglomeration of particles [18].

Energy Dispersive Spectroscopy (EDS) analyses is used to control for the chemical composition of the rubber-based composites as given for the composite n°6 in Figure 3.

In the microstructure, certain aggregation of the particles has been observed. This may be related to the preparation conditions such as milling time and speed. In a general way, there is a homogeneity observed in the distribution of the reinforcement particles in the microstructure. Furthermore, absence of the micro cracks between hard and soft particles indicates that there is a good cohesion among the matrix and the reinforcements in the structure.

In Figure 3, a reference image for the mapping zone is presented and the distribution of the reinforcements in the matrix are given as color presentation. In these mapping pictures, at the left-hand side of the images there is a color legend displays the intensity of identified element that is seen at the right bottom of each image. Upward evolution of the color signifies the increase in the intensity of the specified element. From these figures, homogeneous distribution of the reinforcement elements can be observed. However, casually segregation of the reinforcements is similarly distinguished.

### 3.2. Mechanical Characterization: Three-Point Single-Edge Notched Bending Tests (3PB-SENB)

3P-Bending tests have been carried out with a constant speed of 1 mm/min for the specimens produced from each different type of composites called single-edge notched bending (SENB). The specimens have been tested for each composition and evolution of stress levels depending on the deformation for each composition were presented. Figure 4a shows the finals stage of the powder after 4 h milling of the recycled rubber with reinforcement and hot compacted specimens and finally Three-Point single edge (3PB-SENB) notch bending test device respectively. Figure 4b shows test results for each composition. By using these graphs, fracture toughness and fracture energy values have been calculated.

The graphs in Figure 4b show that the addition of glass bubble microspheres, aluminum chips (Al-chips) and fine gamma alumina fibers (γ-Al2O3) increases the flexural stress of the matrix. Furthermore, the addition of glass bubbles decreases of the strain at break and the density of the composite.

Mechanical and physical properties are obtained by conducting the 3P bending test with single edge notch specimens suggested by ASTM D790 [10]. Flexural stress is calculated during three-point bending as
(1)$$\sigma_{f}=\frac{3\mathrm{PL}}{2{\mathrm{BW}}^{2}}$$
where L is the span length, P is the maximum bending load, and B and W are the sample width and thickness, respectively. Flexural strain is
(2)$$\varepsilon_{f}=\frac{6\mathrm{DW}}{L^{2}}$$
in which D is the maximum deflection at the center of the specimen. The modulus of elasticity in bending is calculated from
(3)$$E_{f}=\frac{L^{3}m}{4{\mathrm{BW}}^{3}}$$
m is the tangent of the initial straight portion of the stress–strain curve.

The critical stress intensity factor in mode I loading is determined by testing of the SENB specimens under plane strain conditions as
(4)$$K_{\mathrm{Ic}}=\frac{P}{B\sqrt{W}}f\left(x\right),\;x=a/W$$

This equation is valid for 0<x<1 where P is the maximum force and “a” is the total notch length. f(x) is the geometry correction factor which is obtained by(5)$$f\left(x\right)=6x^{1/2}\left[\frac{1.99-x\left(1-x\right)(2.15-3.93x+2.7x^{2})}{\left(1+2x\right){\left(1-x\right)}^{3/2}}\right]$$

Critical strain energy release rate (fracture energy) is calculated as
(6)$$G_{\mathrm{Ic}}=\frac{K_{\mathrm{Ic}}^{2}}{E}$$
where E is the modulus of elasticity.

The dimensions of the test specimens are shown in Table 3. Fracture toughness stress intensity factor and critical energy release rate in mode I are calculated and presented in Table 4. Ultimate flexural stress and strain at break values are obtained during the 3P bending test. The rubber-based composites reinforced with glass bubbles and alumina with a certain ratio have shown improvements in the mechanical properties, flexural stress and fracture toughness. Mechanical tests on recycled fresh scrap rubber composites in the literature also show that the reinforcement particles such as alumina fibers, improve the fracture toughness of the composites [22,23,24]. It can be observed from the studies including glass bubble composites in the literature that lower glass bubble content contributes to higher tensile strength [23]. Short glass fibers also improve the mechanical performance of the recycled rubber composites [14]. Then, it is indicated that alumina fibers significantly improve the strain under quasi-static loading and the ability to absorb mechanical energy under static loading [20]. That is why hard microparticles retard the propagation of micro cracks in the structure.

During the tests, crosshead speed is selected between 1–2 mm/min and the change of the properties according to the speed is shown in Table 5 for specimen 3. It can be said that the critical stress intensity factor increases with increasing crosshead speed.

For rubber-based composites, it is also important to consider homogenization process since the matrix and discontinuous fibers have different material characteristics. Among these analytical models, Halpin–Tsai model is convenient to estimate the experimental results in case of homogeneous distribution of the fillers. The Halpin–Tsai model [10] is as follows
(7)$$\frac{E_{c}}{E_{m}}=\frac{1+\xi_{f}{\eta V}_{f}}{1-\xi_{f}{\eta V}_{f}},\;\eta =\frac{\frac{E_{f}}{E_{m}}-1}{\frac{E_{f}}{E_{m}}+\xi },{\;\xi }_{f}=2\frac{L_{f}}{D_{f}}$$
where $E_{c}$, $E_{f}\;$ and $E_{m}$ indicate modulus of elasticity of the composite, fibers and the matrix, respectively. $V_{f}$ and $V_{m}$ refer the volume fraction of the reinforcements and the matrix, respectively; $\xi_{f}$ is the shape factor of the fibers; and $L_{f}$ and $D_{f}$ are the length and diameter of the reinforcement particles, respectively. The classical Halpin–Tsai model is used for composites reinforced with one scale reinforcement, so the modified Halpin–Tsai model is applied to predict the elastic properties of the composites. A modified Halpin–Tsai model for the effects of nano reinforcements is given as
(8)$$E_{c}=\left[\frac{3}{8}\frac{1+\xi_{f}\eta_{L}V_{f}}{1-\eta_{L}V_{f}}+\frac{5}{8}\frac{1+2\eta_{T}V_{f}}{1-\eta_{T}V_{f}}\right]E_{m},{\;\eta }_{L}=\left[\frac{\frac{E_{f}}{E_{m}}-1}{\frac{E_{f}}{E_{m}}+\xi_{f}}\right],{\;\eta }_{T}=\left[\frac{\frac{E_{f}}{E_{m}}-1}{\frac{E_{f}}{E_{m}}+2}\right],\;\xi_{f}=2\frac{L_{f}}{D_{f}}$$

$E_{c}$ is the elasticity modulus of the composites with randomly oriented reinforcements and $E_{f}$ and $E_{m}$ are the elasticity moduli of reinforcements and the matrix. $V_{f}$, $T_{f}$ and $L_{f}$ are the volume fraction, thickness and length of the reinforcement particles, respectively.

For composition 5, there are three different reinforcements which are glass bubbles, alumina fibers and nano SiO_2_, so the modified H-T model should be calculated in three steps. First, the modulus of elasticity of the recycled rubber-epoxy matrix is calculated using the modified H-T equations as follows:(9)$$E_{C1}=\left[\frac{3}{8}\frac{1+\xi_{R}\eta_{L}V_{R}}{1-\eta_{L}V_{R}}+\frac{5}{8}\frac{1+2\eta_{T}V_{R}}{1-\eta_{T}V_{R}}\right]E_{E},{\;\eta }_{L}=\left[\frac{\frac{E_{R}}{E_{E}}-1}{\frac{E_{R}}{E_{E}}+\xi_{R}}\right],{\;\eta }_{T}=\left[\frac{\frac{E_{R}}{E_{E}}-1}{\frac{E_{R}}{E_{E}}+2}\right],\;\xi_{R}=2\frac{L_{R}}{D_{R}}$$
where $E_{C1}$ is the modulus of elasticty of the new matrix; $E_{R}$ is the modulus of elasticity of the recycled rubber; $E_{E}$ is the modulus of elasticity of the epoxy; and $V_{R}$, $D_{R}$, $L_{R}$ and $\xi_{R}$ are the volume fraction, diameter, length, and the shape factor of the recycled rubber, respectively. At that step, the volume fraction and the shape factor of the rubber are 0.783 and 3, respectively. Hereby, the modulus of elasticity of the new matrix has been determined. Then, applying the same formula for the glass bubble reinforced composite
(10)$$E_{C2}=\left[\frac{3}{8}\frac{1+\xi_{G}\eta_{L}V_{G}}{1-\eta_{L}V_{G}}+\frac{5}{8}\frac{1+2\eta_{T}V_{G}}{1-\eta_{T}V_{G}}\right]E_{C1},{\;\eta }_{L}=\left[\frac{\frac{E_{G}}{E_{C1}}-1}{\frac{E_{G}}{E_{C1}}+\xi_{G}}\right],{\;\eta }_{T}=\left[\frac{\frac{E_{G}}{E_{C1}}-1}{\frac{E_{G}}{E_{C1}}+2}\right],\xi_{R}=2\frac{L_{G}}{D_{G}}$$
in which $E_{C2}$ is the modulus of elasticty of the new matrix; $E_{G}$ is the modulus of elasticty of glass bubble particles; and $V_{G}$, $D_{G}$, $L_{G}$ and $\xi_{G}$ are the geometric properties of the glass bubble particles. Here, the volume fraction and the shape factor of the glass bubble microspheres are 0.547 and 2, respectively. Adding the alumina fibers to the composite
(11)$$E_{C3}=\left[\frac{3}{8}\frac{1+\xi_{\mathrm{AF}}\eta_{L}V_{\mathrm{AF}}}{1-\eta_{L}V_{\mathrm{AF}}}+\frac{5}{8}\frac{1+2\eta_{T}V_{\mathrm{AF}}}{1-\eta_{T}V_{\mathrm{AF}}}\right]E_{C2},{\;\eta }_{L}=\left[\frac{\frac{E_{\mathrm{AF}}}{E_{C2}}-1}{\frac{E_{\mathrm{AF}}}{E_{C2}}+\xi_{\mathrm{AF}}}\right],{\;\eta }_{T}=\left[\frac{\frac{E_{\mathrm{AF}}}{E_{C2}}-1}{\frac{E_{\mathrm{AF}}}{E_{C2}}+2}\right],\xi_{\mathrm{AF}}=2\frac{L_{\mathrm{AF}}}{D_{\mathrm{AF}}}$$

Here, $E_{C3}$ is the modulus of elasticity of the matrix; $E_{\mathrm{AF}}$ is the modulus of elasticty of the alumina fibers; $E_{C3}$ is the modulus of elasticity of the matrix; and $E_{\mathrm{AF}},{\;V}_{\mathrm{AF}}$, $D_{\mathrm{AF}}$, $L_{\mathrm{AF}}$ and $\xi_{\mathrm{AF}}$ are the modulus of elasticity, the volume fraction, diameter, length and the shape factor of the alumina fibers, respectively. The volume fraction of alumina fiber is 0.018 and the shape factor is 3.33. Finally, the modulus of elasticity of the composite is obtained by adding nano SiO_2_ particles as
(12)$$E_{C}=\left[\frac{3}{8}\frac{1+\xi_{{\mathrm{SiO}}_{2}}\eta_{L}V_{{\mathrm{SiO}}_{2}}}{1-\eta_{L}V_{{\mathrm{SiO}}_{2}}}+\frac{5}{8}\frac{1+2\eta_{T}V_{{\mathrm{SiO}}_{2}}}{1-\eta_{T}V_{{\mathrm{SiO}}_{2}}}\right]E_{C3},{\;\eta }_{L}=\left[\frac{\frac{E_{{\mathrm{SiO}}_{2}}}{E_{C3}}-1}{\frac{E_{{\mathrm{SiO}}_{2}}}{E_{C3}}+\xi_{{\mathrm{SiO}}_{2}}}\right],\;\;\eta_{T}=\left[\frac{\frac{E_{{\mathrm{SiO}}_{2}}}{E_{3}}-1}{\frac{E_{{\mathrm{SiO}}_{2}}}{E_{3}}+2}\right],\xi_{{\mathrm{SiO}}_{2}}=2\frac{L_{{\mathrm{SiO}}_{2}}}{D_{{\mathrm{SiO}}_{2}}}$$
where $E_{C}$ is the modulus of elasticity of the recycled rubber-based composite reinforced with the glass bubble, alumina fiber and SiO_2_ particles; 0.013 is the volume fraction; and 1.0 is the shape factor of silica particles.

The same procedure is carried out for all of the composites and the results are presented in Table 6 comparing with the experimentally obtained moduli.

Although the elastic moduli calculated from the experimental and Halpin–Tsai model are not close to each other, it can be stated that the general trend of change in elastic moduli are similar according to the composition of the composites. For instance, the increasing content of the glass bubble and fiber particles increases the modulus of elasticity of the composite. In addition, it is seen that increasing the weight fraction of alumina fibers causes an increase in elastic modulus values, both experimentally calculated and calculated with the Halpin–Tsai model. In former studies [6], it has been observed that modulus of elasticity of rubber-based composites increases numerically by adding reinforcements due to high modulus of fibers.

Furthermore, fracture toughness measurements from 3P bending test are simulated. The elastic stress analysis is performed using ABAQUS/Implicit. Finite element model representing the dimensions used in experiments is performed to calculate the maximum energy release rate of each specimen. Symmetric boundary conditions are applied along the corresponding edges of symmetry. Figure 5 shows the typical configuration and mesh of one of the models for plane strain condition. In order to obtain high stress field around the crack, relatively fine mesh sizes are adopted in this region as seen in Figure 6. Material properties obtained from the test result are used in the calculations. In linear elastic fracture mechanics, the stress intensity factor is used to characterize the fracture toughness of a brittle material. The fracture toughness is assumed as constant for a given specimen thickness. To check the validity of this assumption regarding the specimen configuration, the stress intensity factors are calculated for the maximum deflections observed from bending tests. To implement the test result, the displacements are given at the center of the specimen and elasticity modulus is used as in the experimental results. After that, maximum force is obtained and compared with the experimental value. The proportion of these two values is used to modify the elasticity modulus. After modifying the elasticity modulus, the model is run again, and elasticity modulus modification is done until a coherent maximum force regarding experimental results is obtained. After converging to the maximum force, energy releasing rate of the composites is determined along the specimen thickness. Mechanical properties are implemented for finite element analysis for specimens 2, 3 and 4 and the critical stress intensity factors are calculated very close to the values obtained by experimental values and shown in Table 7.

### 3.3. Fracture Surface Analyses of the Composites; Damage Mechanisms

Fracture surfaces of all of the compositions studied here were presented in the Figure 7. At the first observation, quasi all of the compositions have shown that there is a strong cohesion between the reinforcements and matrix. It means that the manufacturing parameters were well chosen according to the devulcanization process of the recycled rubber and molar ratio of each reinforcement in the matrix.

Fracture surfaces analyses were carried out by means of SEM to identify the toughening mechanisms. First, a good cohesion is observed between matrix and reinforcements. In addition, a few fracture deviations are also perceived because of the effect of added reinforcements.

A few micro cracks observed that they weekly propagate just after disappear. This case is explained due to the crack pinning for the toughening of the composites. All of these cases should be explained with a good chemical bonding diffusion during the compaction. We supposed that the devulcanization process of the recycled rubber carried out by using microwave treatment are very helpful to encourage of the chemical bonding diffusion. Because, during the vulcanization free-links of the virgin rubbers are linked with sulfur atoms to improve the material properties [6]. Then, it becomes difficult to regenerate chemical bonds between rubber, reinforcement and epoxy. However, in this study, free links on the rubbers were regenerated at the level of 10–12% by means of pre-treatment of the recycled rubber in order to improve the toughness of the composites designed here.

## 4. Conclusions

Recycled rubber-based composites have been reinforced with glass bubble microspheres, short glass fibers, aluminum chips (Al-chips) and fine gamma alumina fibers (γ-Al_2_O_3_) reinforcements containing minor reinforcements such as nano silica (SiO_2_) and boron nitride (BN) to provide multifunctionality to the composites. Glass bubbles are preferred as one of the major reinforcements because of their perfect weight reduction, performance, processing and dimensional stability characteristics [15]. Then, alumina fibers are used as they have high compressive strength as well as good thermal conductivity [16,17]. A special process has been applied to complete the manufacturing of the composites after determination of the reinforcements. This process is indicated in the former works [6,7,8,19,20,21]. Toughening mechanisms have been analyzed and mechanical and physical properties (fracture toughness stress intensity factor and critical energy release rate) have been determined by fracture toughness tests. Microstructural and fracture surfaces analysis have been carried out by means of scanning electron microscopy. After fracture surfaces have been obtained from 3 PB tests and analyzed by means of SEM, a good adhesion of the reinforcements has been observed by creating ideal interface for each composition. It is observed that the rubber based composites reinforced with glass bubbles, glass fibers and alumina fibers have shown improvements in the mechanical properties, flexural stress and fracture toughness. The improved toughening mechanisms are due to the formation of the cavitation and void formation because of debonding of the fine reinforcement particles in the rubber matrix which leads to void growth with consequent locally matrix expansion. For this reason, the mixture of recycled rubber (80 wt%) with solid powder epoxy resin (20 wt%) is significant for manufacturing the effective composite material. By using various characterization methods, the physical and mechanical properties of these composites are determined, and they offer an effective mechanical performance based on strength, elasticity modulus and strain values. It is also observed from numerical analysis that the stress intensity factor is influenced by the depth and crack length of the specimen. This study can be useful in terms of providing an idea that fresh scrap rubber components to be used in intermediate and advanced applications can reduce costs in various applications and CO_2_ emissions due to their lightweight properties. In the near future, we will carry out a comprehensive study with mathematical modelling by using the representative volume element (RVE) model that will be presented damage behavior under very high strain rates applications.
**ANNEXE****DETAIL INFORMATION ON THE MATERIALS/CHEMICAL PRODUCTS IN THE PRESENT WORK****Glass Bubble Microspheres**Glass bubbles K1 taken from 3M are hollow glass microspheres (diameter variable 2–50 µm). They have a very low typical density (0.125 g/cc) with a stable viscosity. They give a good strength to density ratio. They are non-flammable and non-porous with their low alkalinity; for this reason, they are compatible with most resins while providing do not absorb the resin.**Pure Dry Epoxy Resin**Produced by a Chinese Society, High Purity Solid Epoxy Resin. NEWPOX-0212 with E12 Grade Epoxy Resin based on Bisphenol-A. Epoxide Equivalent Weight (780 g/equivalent) and Softening Point (95 °C). It has very good cohesion with rubber powder.**The following materials were received by WDR Chemical Society**Short glass fibers have a density of 2.45 g/cm^3^, softening temperature is found around 1000 °C and Thermal expansion is about at the level of 2.85 µm/m °CFine pure aluminium chips (AA1050-chips), with a density of 2.7 g/cm” as thin leaves with a mean size of 100–150 µmNano Silica (SiO_2_) with a surface area 180–200 m^2^/g, it is a white soft powder composed of high purity amorphous silica powderGamma alumina fibers (γ-Al_2_O_3_) diameter: 10 µm and length: 25–40 µmHexagonal-Boron Nitride (hBN) with a density of 2 g/cm^3^. It has an excellent thermal and chemical stability, very easily mixture in rubbers, it has self-lubricating properties**Fresh scrap recycled rubber (NBR)**A rubber society in Sofia Bulgaria produce some special pieces, etc. on the production line, many fresh scrap sheets and as soon as they grind these scraps in the factory. We receive it as a granular form with a grain size variable 50–100 µm by means of UTCM university of Chemical Technology and Metallurgy where we collaborate for a long timeWe heat it 4 min in the microwave before silanization process after that we mix with fine epoxy powder and also with different reinforcement. Mainly this chemical treatment was carried out in the E. Chemistry laboratory in CNAM-Arts et Métiers Paris. Finally we make milling all the mixture during 4 h before hot compaction. In fact, during this process, sulfur links are tried to be broken and partially new other links are generated, then the flowing capacity and interaction of recycled rubbers with other substances are increased. In the literature, some works have already indicated that devulcanization cannot be achieved completely. As we have shown in our former works, we have only carried out 10–12% of devulcanization. After determination of the reinforcements with matrix in wt%, a special process has been applied to complete successfully the manufacturing of the composites. This process involves silanization of the recycled rubber and devulcanization before blending it with pure dry-epoxy resin and reinforcements. With this simple process, we try to improve only partially good liaison between recycled rubber matrix with the epoxy resin.

## Figures and Tables

**Figure 1 polymers-13-04215-f001:**
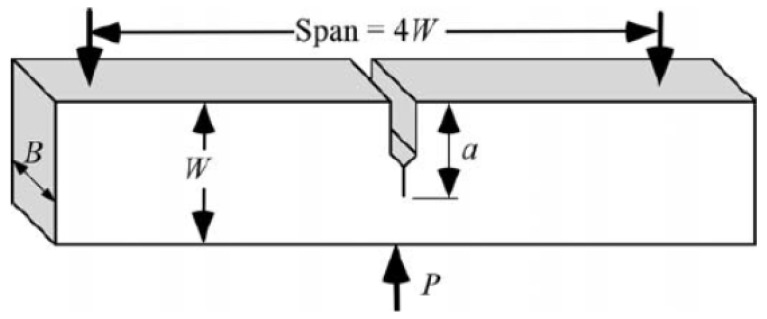
Three-Point single edge notch bending (SENB) test for fracture toughness, K_IC_ and fracture energy, G_IC_ characterization.

**Figure 2 polymers-13-04215-f002:**
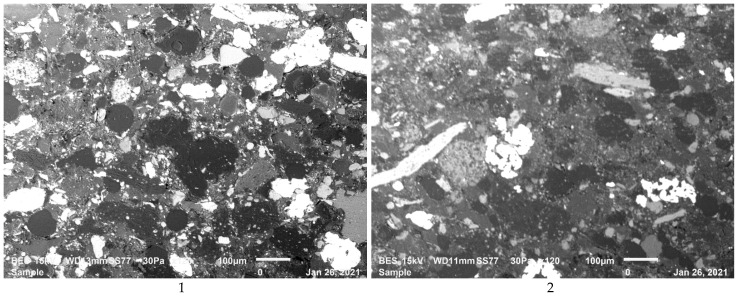
Microstructures of the composites studied here from the composite (**1**–**7**), respectively, after sectioning and polishing.

**Figure 3 polymers-13-04215-f003:**
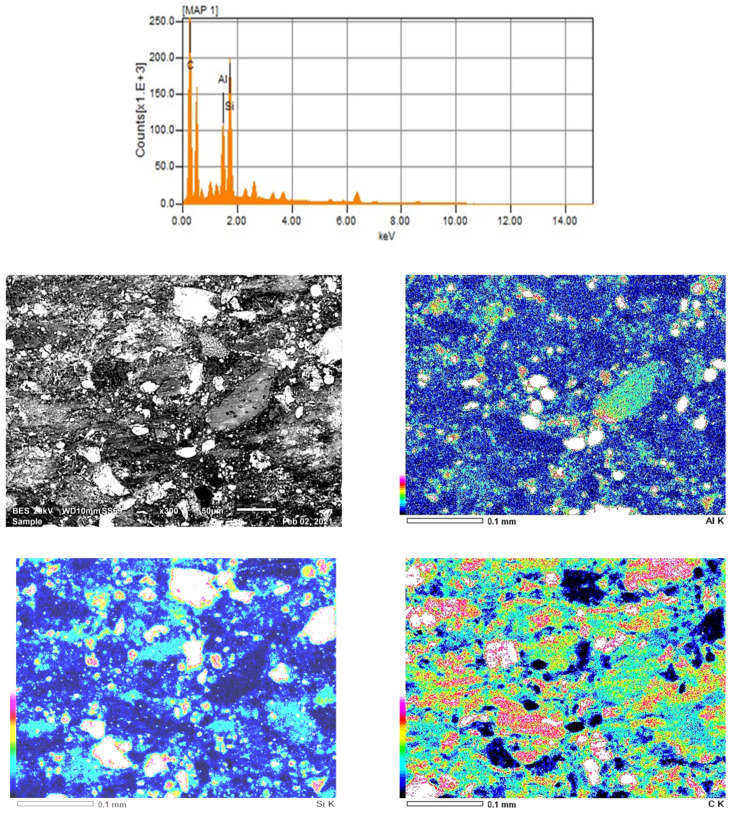
Energy Dispersive Spectroscopy (EDS) Chemical and Mapping for Specimen 6.

**Figure 4 polymers-13-04215-f004:**
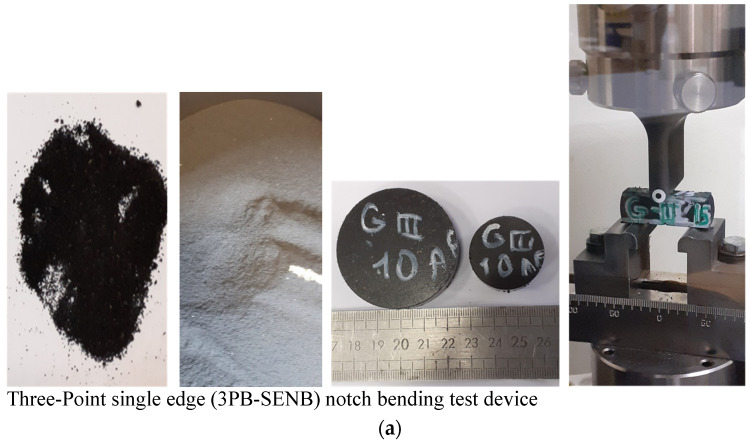
(**a**) Final milling mixture of recycled rubber and reinforcements and hot compaction of the specimens and (**b**) Three-Point single edge notch bending test results (3PB-SENB) for the different compositions.

**Figure 5 polymers-13-04215-f005:**
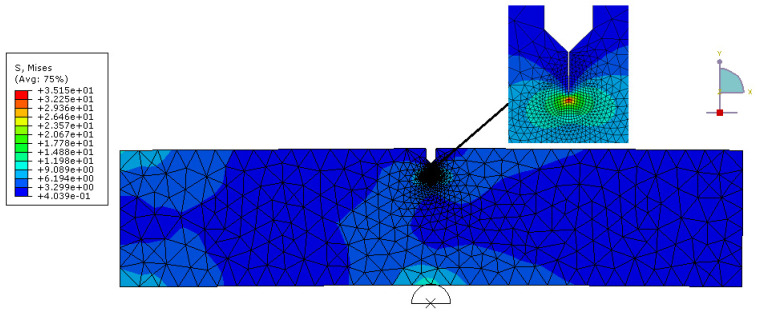
Numerical Model for 3P single edge notch bending (SENB) test for composite 4.

**Figure 6 polymers-13-04215-f006:**
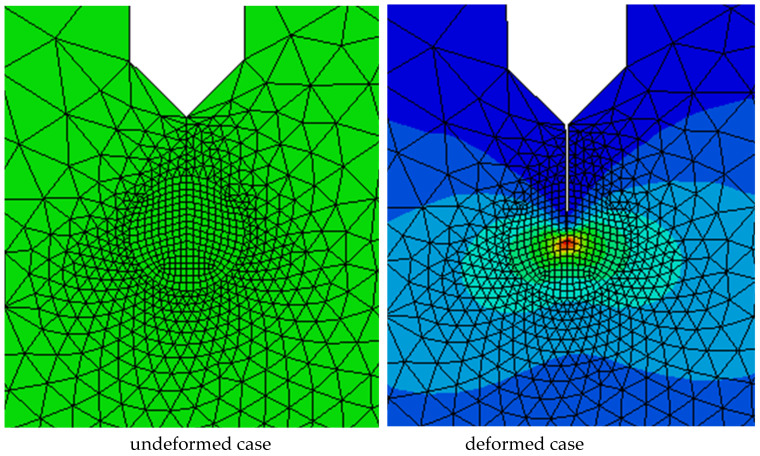
Finite element mesh according to the geometry given in the Figure 5.

**Figure 7 polymers-13-04215-f007:**
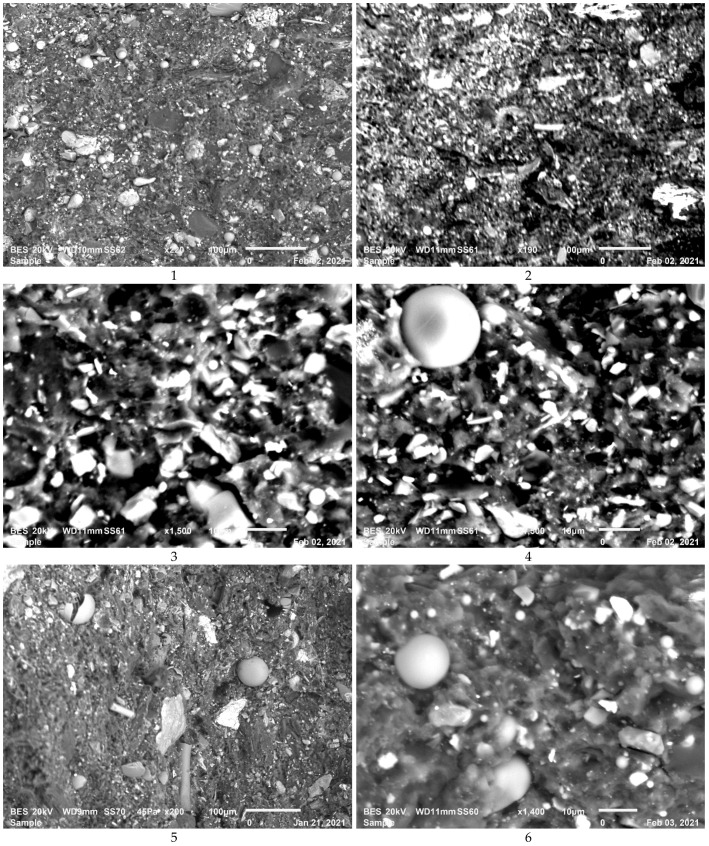
(**1**–**7**) Fracture Surfaces of the seven Composites failed after 3PB–SENB Tests.

**Table 1 polymers-13-04215-t001:** Chemical compositions of the composites studied in this work (wt%).

Composite No	Matrix (Rubber/Epoxy)	Glass Bubble	Glass Fiber	γ-Al_2_O_3_ Fiber	γ-Al_2_O_3_ Sphere	Al-Chips	Nano SiO_2_	BN
1	95 (80/20)					5		
2	90 (80/20)					10		
3	80 (80/20)					20		
4	90 (80/20)	5		5				
5	75 (80/20)	10		10			5	
6	85 (80/20)				10			5
7	75 (80/20)		15				10	

**Table 2 polymers-13-04215-t002:** Shore D hardness measurements of the composites.

Composite Number	SHORE-D TEST ASTM 2240
1	68 ± 2
2	72 ± 2
3	80 ± 3
4	81 ± 2
5	80 ± 3
6	79 ± 2
7	80 ± 3

**Table 3 polymers-13-04215-t003:** Dimensions of the specimens.

Composition Number	Span Length	Width	Thickness	Total Notch Length
	L (mm)	W (mm)	B (mm)	a (mm)
1	32	8	15	1.25
2	32	7.87	9.12	1.25
3	32	7.18	20.8	1.25
4	32	7	19	1.25
5	32	6.45	18	1.25
6	32	7.66	17.33	1.25
7	32	6.31	19	1.25

**Table 4 polymers-13-04215-t004:** Comparison of mechanical properties of the specimens.

Composition Number	Ultimate Flexural Stress (MPa)	Strain at Break	Modulus of Elasticity (MPa)	Geometry Correction Factor	Critical Stress Intensity Factor	Critical Strain Energy Release Rate
	$\sigma_{fb}$	$\varepsilon_{f}$	E (MPa)	f (a/W)	$K_{Ic}$ $\left(MPa.m^{1/2}\right)$	$G_{Ic}$ $\left(kJ/m^{2}\right)$
1	12.55	0.0357	940.0	4.15	0.78	0.64
2	29.19	0.0226	1214.0	4.18	1.77	2.59
3	44.27	0.0237	1903.0	4.37	2.45	3.16
4	34.97	0.0618	1914.0	4.43	1.89	1.87
5	59.93	0.0149	2784.0	4.62	2.99	3.21
6	28.99	0.0142	1289.5	4.24	1.71	2.28
7	17.56	1.1923	1827.0	4.68	0.86	0.40

**Table 5 polymers-13-04215-t005:** The effect of speed on $K_{\mathrm{Ic}}$ and $G_{\mathrm{Ic}}$ (kJ/$m^{2}$ ) values.

Composition Number	Crosshead Speed		Geometry Correction Factor	Critical Stress Intensity Factor	Critical Strain Energy Release Rate
	v (mm/min)	a/W	f (a/W)	$K_{Ic}$ $\left(MPa.m^{1/2}\right)$	$G_{Ic}$ $\left(kJ/m^{2}\right)$
3-1	1.5	0.17	4.38	2.45	3.16
3-2	2	0.17	4.38	2.85	4.27

**Table 6 polymers-13-04215-t006:** Modulus of elasticity obtained by Halpin–Tsai Homogenization.

Composite Number	Density of the Composite ($g/cm^{3}$)	Halpin–Tsai Modulus $E_{c}$ (MPa)	Experimental Modulus E (MPa)
1	1.118	710.70	940.00
2	1.151	754.26	1214.00
3	1.225	857.47	1903.00
4	0.990	1219.16	1914.00
5	0.932	1717.83	2784.00
6	1.149	977.17	1289.50
7	1.260	1057.01	1827.00

**Table 7 polymers-13-04215-t007:** Critical Stress Intensity Factors obtained by Numerical Analysis.

Composition Number	Ultimate Flexural Stress (MPa)	Maximum Deflection at Center	Modulus of Elasticity	Crack Length/Width	Experimental	Numerical Analysis
	σ_fB_	D (mm)	E (MPa)	a/W	$K_{Ic}$ $\left(MPa.m^{1/2}\right)$	$K_{Ic}$ $\left(MPa.m^{1/2}\right)$
2	29.19	0.49	1214	0.16	1.77	1.96
3	44.27	0.56	1903	0.17	2.45	2.21
4	34.97	1.51	1914	0.18	1.89	1.88

## Data Availability

Not applicable.

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
