# Peer review of "Toughening Mechanism Analysis of Recycled Rubber-Based Composites Reinforced with Glass Bubbles, Glass Fibers and Alumina Fibers"

_polymers, 2021, doi:10.3390/polym13234215_

Round 1

Reviewer 1 Report

The authors revised their manuscript according to my suggestions. Thus the manuscript can be accepted for publication

Author Response

 Dear Reviewer

Thank you for your kind valuable review again and second time youhave spent your time.

We really appreciated your suggestions and critics
Kind Regards

E. BAYRAKTAR

Reviewer 2 Report

Introduction should be enhanced. Also it should also refer to the other works than those of Authors. Generally, the amount of self-citations is too high, 10/21.

All materials should be listed in the manuscript with their origin and properties, so the work would be repeatable for others. 

Citations should be ordered. Currently, in text 19-21 is just after 11.

Please present the details of rubber particles applied. How they were generated especially. What type of shredding process was applied. Presenting SEM image would be very helpful.

"In this study, glass bubbles are preferred as major reinforcement" - but Al-chips were analyzed in more compositions, so it is not true.

Generally how were the compositions selected? Why for example there is no combination of glass bubbles with boron nitride?

"recycled rubbers do not carry any free chains to form new bonding with epoxy resin because of former vulcanization process" - what about chain scission ocurring during shredding of rubber?

Please describe the applied devulcanizaiton process. What type of the process was applied? What additional compounds? What conditions?

Please describe the applied silaniation process. What compounds were applied? What conditions?

Please present the comparison of rubber characteristics before the modification, after devulcanization and after silanization.

"Chemical treated followed devulcanized recycled rubber have been mixed with epoxy resin for obtaining a strong chemical bonding diffusion reinforced with different reinforcement." - please provide any results confirming the strong chemical bonding.

All tests should be described in the experimental section, so the information or hardness test applied should be given.

Why such a high amount of Si was detected in sample 6 and there is no indication of boron or nitrogen?

Information about calculation of flexural strength should be presented in the experimental section.

Same for dimensions of specimens.

Considering the calculations based on Halpin-Tsai model, please provide all parameters for all fillers, so present the shape factors of them and volume fractions in composites. It should be all given in the experimental section.

What are the magnifications of SEM images? The information is hardly visible on most of images.

Discussion of all results must be enhanced with the referring to other literature works dealing with rubber composites.

Author Response

Thank you for your suggestions and valuable constructible critics.

In general way, we have made a considerable revision an improved our manuscript according to the suggestions and critics of the reviewers

All materials should be listed in the manuscript with their origin and properties, so the work would be repeatable for others. 

In fact, we work with recycled rubber more than 10 years. In fact, a rubber society produce some sport wears etc. on the production line, many fresh scrap sheets and as soon as they grind these scraps in the factory. We receive it as a granular form with a grin size 100-150µm.

We heat it 4 minutes in the microwave before silanization process after that we mix with fine epoxy powder and also with different reinforcement. Mainly this chemical treatment was carried out in the E; Chemistry laboratory in CNAM-Arts et Métiers Paris. Finally we make milling all the mixture during 4 hours before hot compaction. We did not repeat these operations again in this work for not increasing the similarities of the text, because, we have written many times in former publications.

In fact, during this process, sulphur links are tried to be broken and partially new other links are generated, then the flowing capacity and interaction of recycled rubbers with other substances are increased. In the literature, some works have already indicated that devulcanization cannot be achieved completely. As we have shown in our former works, we have only carried out 10-12% of devulcanization. After determination of the reinforcements with matrix in wt%, a special process has been applied to complete successfully the manufacturing of the composites. This process involves silanization of the recycled rubber and devulcanization before blending it with epoxy resin and reinforcement. With this simple process, we try to improve only partially good liaison between recycled rubber matrix with the reinforcement.

Reviewer 3 Report

 Toughening Mechanism Analysis of Recycled Rubber Based Composites Reinforced with Glass Bubbles, Glass Fibers and Alumina Fibers

The paper seems to be interesting but some changes are required before final acceptance:

Comments:

  1. In the Introduction, the author described past work, but little comment on the contribution and shortcomings. The author needs to provide critical comments.
  2. Explore more applications and add them to the introduction.
  3. How did you get table 1?
  4. More physical insight into the Discussion section is needed.
  5. In the conclusion, please show how the work advances the field from the present state of knowledge. Please provide a clear justification for your work in this section, and indicate uses and extensions if appropriate. Moreover, you can suggest future experiments/simulations and point out those that are underway.

Author Response

Thank you for your suggestions and valuable constructible critics.

In general way, we have made a considerable revision an improved our manuscript according to the suggestions and critics of the reviewers

In fact, this research project is going on with aeronautical applications under the high strain rate conditions. We did not give (dare) some of the details at this stage. Many detail information were given in our former publications on the production process and we will develop this project in the way of a comprehensive study with mathematical modelling by using the representative volume element (RVE) model that will be presented damage behaviour under very high strain rates applications.

Round 2

Reviewer 1 Report

The paper presents a series of information on certain properties of rubber matrix composites.

- The paper, through content, could be of interest to the scientific community:

- no abbreviations are allowed in the keywords;

- The introduction needs to be substantially improved and other information related to the research presented needs to be added;

- the research methodology must be presented more clearly. It is necessary to justify the decision to choose a certain composition for the 7 samples;

- a better characterization of the materials and in particular of the devulcanised rubber used is required. The properties of this rubber differ substantially depending on the parameters chosen for the devulcanization technology, but also on the characteristics of the recycled rubber waste;

- Macroscopic images of the specimens made must be presented;

- Most SEM images have an inadequate resolution and are not explained in the figure caption;

- The FEM method applied using ABAQUS / Implicit is summarized and must be detailed considering that it refers to composite materials;

- discussions need to be more detailed in order to highlight the contributions of research;

- the conclusions should be more concrete and include future research directions.

Author Response

Thank you for your valuable contributions and suggestions with your critics.

We have completely revised and improved our manuscript by this occasion. We have corrected and explained with more information about the characterization of the materials and in particular of the recycled rubber that we used.

In the present paper, we tried to heat with microwave oven during 4 minutes to be devulcanized it. During this process, sulphur links are tried to be broken and partially new other links are generated, then the flowing capacity and interaction of recycled rubbers with other substances are increased. In the literature, some works have already indicated that devulcanization cannot be achieved completely. As we have shown in our former works, we have only carried out 10-12% of devulcanization. After determination of the reinforcements with matrix in wt%, a special process has been applied to complete successfully the manufacturing of the composites. This process involves silanization of the recycled rubber and devulcanization before blending it with epoxy resin and reinforcement. With this simple process, we try to improve only partially good liaison between recycled rubber matrix with the reinforcement.

Actually we have rewritten the conclusion with more compact information on the research that is going on.

Reviewer 2 Report

Some comments from previous review were not adressed:

"Citations should be ordered. Currently, in text 19-21 is just after 11" - Citations are still not in order, after 8 is 17-22, then 13. 

"All materials should be listed in the manuscript with their origin and properties, so the work would be repeatable for others. " - origin and properties still not provided.

"Please present the details of rubber particles applied. How they were generated especially. What type of shredding process was applied. Presenting SEM image would be very helpful" - no new information about rubber particles was provided.

""In this study, glass bubbles are preferred as major reinforcement" - but Al-chips were analyzed in more compositions, so it is not true." - comment not adressed.

"Generally how were the compositions selected? Why for example there is no combination of glass bubbles with boron nitride?" - comment not adressed.

""recycled rubbers do not carry any free chains to form new bonding with epoxy resin because of former vulcanization process" - what about chain scission ocurring during shredding of rubber?"  - comment not adressed.

"Please describe the applied silaniation process. What compounds were applied? What conditions?"  - comment not adressed.

"All tests should be described in the experimental section, so the information or hardness test applied should be given."  - comment not adressed.

"Why such a high amount of Si was detected in sample 6 and there is no indication of boron or nitrogen?"  - comment not adressed.

"Considering the calculations based on Halpin-Tsai model, please provide all parameters for all fillers, so present the shape factors of them and volume fractions in composites. It should be all given in the experimental section." - shape factor for silica particles is precisely 1.0, so all of them are ideal spheres? Please provide SEM images of filler. Moreover, there is no data for glass fiber, Al2O3 spheres, Al chips and boron nitride. It should be presented.

"What are the magnifications of SEM images? The information is hardly visible on most of images." - comment not adressed.

"Discussion of all results must be enhanced with the referring to other literature works dealing with rubber composites." - comment not adressed.
